# Static Temperature Guideline for Comparative Testing of Sorption Heat Storage Systems for Building Application

**Benjamin Fumey [1,\*] and Luca Baldini [2]**

1    Empa Swiss Federal Laboratories for Materials Science and Technology, Überlandstrasse 129,
     8600 Dübendorf, Switzerland
2    Zurich University of Applied Sciences, School of Architecture, Design and Civil Engineering,
     8401 Winterthur, Switzerland; luca.baldini@zhaw.ch
\*    Correspondence: benjamin.fumey@empa.ch; Tel.: +41-58-765-47-84

**Abstract:** Sorption heat storage system performance heavily depends on the operating temperature. It is found that testing temperatures reported in literature vary widely. In respect to the building application for space heating, reported testing temperatures are often outside of application scope and at times even incomplete. This has led to application performance overestimation and prevents sound comparison between reports. This issue is addressed in this paper and a remedy pursued by proposing a static temperature and vapor pressure-based testing guideline for building-integrated sorption heat storage systems. By following this guideline, comparable testing results in respect to temperature gain, power and energy density will be possible, in turn providing a measure for evaluation of progress.

**Keywords:** sorption thermal energy storage; building application; static testing guideline; uniform performance evaluation; space heating





## 1. Introduction

To increase the use of renewable energy, long-term storage systems, with storage periods ranging from weeks to months, are a potential key player. Due to the long storage period, it is in the nature of this application to have a strongly reduced number of charging and discharging cycles compared to a diurnal storage system. This greatly increases the cost per stored energy and makes economic viability challenging [1]. Thermal storage systems, releasing energy in the form of heat, may meet the necessity for low storage material cost more readily than electro-chemical technologies. Proposed for this purpose are sorption technologies, addressing adsorption, liquid absorption and solid absorption [2]. Sorption heat storage systems typically experience heat loss only in the process of charging and discharging, facilitating lossless storage over time [2]. In the sorption heat storage process, serviceable heat is released from ambient gains through sorption (chemically-driven heat pump process) and by exothermic reaction [2–9]. A typical application proposed is domestic space heating.

In the research community it is recognised that the storage material and component interaction can be deciding for the technical feasibility. There is a call for more targeted component development responding to the respective material development. It is found that components, particularly the heat and mass exchanger, are less understood than initially thought. In order to obtain a better comparison between system performance of different technologies, a standardized and simplified testing procedure under realistic operation conditions, i.e., temperatures, is required [10].

This work pursues to answer this call for the building space heating application. A description of the diverse sorption heat storage operating principle is followed by a short exploration into literature, pointing to the absence of sufficient comparable data, frequent

deficient data and the need for a uniform testing procedure. In a final step, a realistic testing procedure for the sorption heat storage system for space heating is proposed.

## 2. Operating Principle

The sorption heat storage system, a type of heat transformer with sorption storage, functions on the principle of sorbate release and uptake on a sorbent, based on the temperature, pressure and sorbent concentration (mass fraction) equilibrium. This equilibrium is most typically altered by temperature-swing-process. Heating the combined sorbent and sorbate kindles sorbate release as concentration equilibrium state is surpassed due to the temperature increase. Exposure of concentrated sorbent, at super equilibrium state (low sorbate content in respect to sorbent temperature) to corresponding elevated sorbate vapor pressure, prompts sorbate uptake, yielding heat release from sorbate condensation and mixing with sorbent.

The basic theoretical performance of a sorbent working pair (sorbent and sorbate) is evaluated by means of vapor pressure vs. temperature diagram in respect to concentration (mass fraction). Figure 1 shows this diagram for aqueous sodium hydroxide (NaOH). Based on the working pair concentration, and the sorbate vapor pressure, the resulting equilibrium concentration can be derived. The diagram shows the charging and discharging process, both clearly dependent on the absorbate vapor ($H_2O$) pressure. In the example, the condensing temperature in the charging process is taken to be 38 °C, resulting in a vapor pressure of 6.6 kPa (see $H_2O$ vapor curve). Under these conditions, heating the sorbent to 80 °C will result in an equilibrium concentration of 50 wt % NaOH. Lower condensing temperatures, i.e., lower absorbate vapor pressure, will increase the equilibrium sorbent concentration as will greater sorbent temperatures, and vice versa. In discharging, the same process holds. Taking the evaporating temperature to be 1 °C, as in the figure, a sorbate vapor pressure of 0.65 kPa results. Under these conditions and with a sorbent concentration of 50 wt %, a maximum equilibrium temperature of 38 °C can be achieved. If the evaporating temperature is increased, then the maximum output temperature also increases. In discharging, the minimum sorbent temperature is significant. This defines the minimum NaOH equilibrium concentration, in turn determining the system energy density, based on the concentration difference between the charged and discharged sorbent solution. In the example, the minimum absorber temperature is taken to be 28 °C, thus reaching a minimum final concentration of 43 wt %.

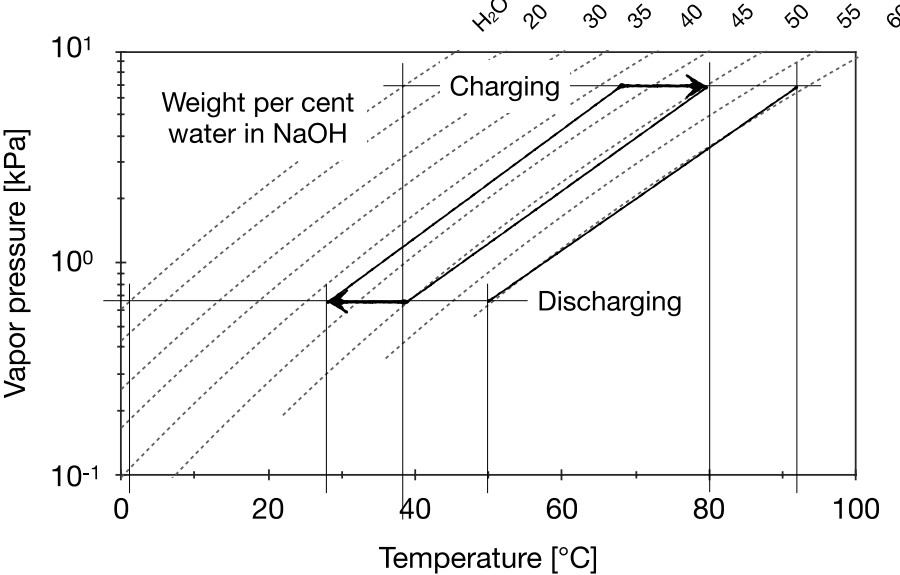

**Figure 1.** Vapor pressure vs. temperature diagram of varying aqueous sodium hydroxide concentrations [11].

If the minimum temperature is lower, or the evaporating temperature higher, the final concentration is reduced and the energy density of the storage increased. This example shows that in charging, the concentration depends on the condensing and the maximum desorbing temperature and in discharging it depends on the evaporating and the minimum absorbing temperature. Additionally, the maximum temperature in discharging is essential, deciding if the system is able to reach the required output temperature under the given operating condition (evaporating temperature). Thus, on a material level, any declaration of energy density must be accompanied by five material temperatures, that is the sorbate temperature and the condensing temperature in charging and the evaporating temperature, and the maximum and minimum sorbent temperatures in discharging.

In the technical realisation of sorption heat storage systems, four process varieties are followed. These are; closed transported, closed fixed, open transported and open fixed [12]. Figure 2 illustrates charging and discharging modes with system boundary interaction, heat transfer fluid (HTF) temperature and vapor pressure of the variations. In the open process heat and mass are released to the ambient air. In the closed approach sorbate is contained and only heat released. In the fixed process sorbent is stationary, in the transported it is mobile.

Closed processes require heat and mass exchangers (HMX) for sorption and desorption and evaporation and condensation and operation is performed under sorbate vapor atmosphere. The process is governed by the respective sorbent and sorbate temperatures, sequel to the corresponding HTF input and output temperatures and depending on the sorbent and HMX design. In charging, evaporation of sorbate from the sorbent is attained by high temperature heat supply to the desorber and condensation of sorbate is acquired by low temperature heat sink on the condenser. State of charge depends on temperature disparity between desorption and condensation and deviation from sorbent equilibrium concentration. Discharging follows the reverse process by supply from low temperature heat source for evaporation, and heat rejection to a middle temperature heat sink on the sorber. Temperature difference and kinetics (state of equilibrium) govern the process.

Open systems join heat and mass transport by air stream and operate at practically ambient pressure. In charging, hot air (heat supply) is blown through the sorbent in the desorber, releasing heat to the sorbent for sorbate evaporation and ejecting vapor to the ambient (vapor sink). The process is managed by the temperature and vapor pressure of the input air, the output air temperature and exposure time. The converse takes place in discharging, cold humid air (vapor source) is delivered to the sorbent, vapor is taken up by the sorbent and heat is released to the heat sink. Again, the process is governed by the input air temperature and vapor pressure as well as output air temperature.

There are several variations in the design of the open systems, frequently accompanied by an air to air heat exchanger. Figure 3 shows variations thereof, with heat release by liquid HTF, as in closed systems, on the left and air-bound heat release and heat recovery on the right. Figure 4 shows an illustration of an open transported system with sorbent to air semi counter flow, air to water heat exchanger and air to air heat exchanger from ITW University of Stuttgart [10]. In spite of these variations, heat source and sink requirements as shown in Figure 2 remain unchanged.

There is a further concept in the open system, where water vapor is sourced from the buildings eject air. Such a system is proposed for example by Tatsidjodoung et al. [13] and Weber et al. [14], shown in Figure 5. Since this approach sources water vapor (latent energy) from the building internally and not externally, no energy is gained and no net heating arrived at. The system functions as an optimized heat exchanger with temperature upgrade. This is addressed by authors such as Gaeini et al. [15] In their open process, ambient air is passed through a ground source heat exchanger and humidified, in order to source heat from the ground, as shown in Figure 6. Consequently, all sorption heat storage processes depend on high temperature heat supply and ambient heat sink in charging and ambient heat source and heat sink in discharging.

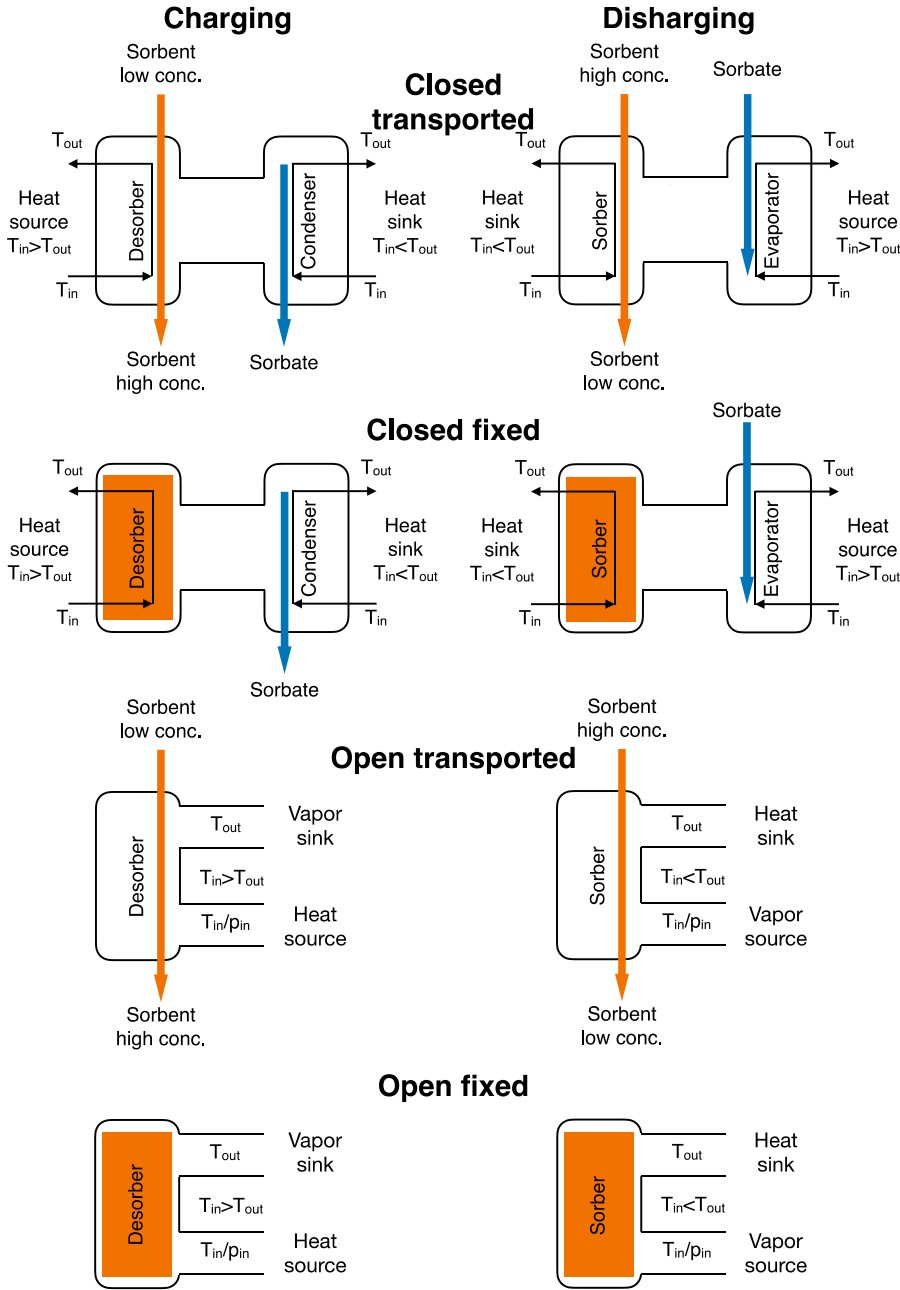

**Figure 2.** Illustrations of the four processes in charging and discharging. Indicated are the relevant heat source and sink temperatures and vapor pressures [12].

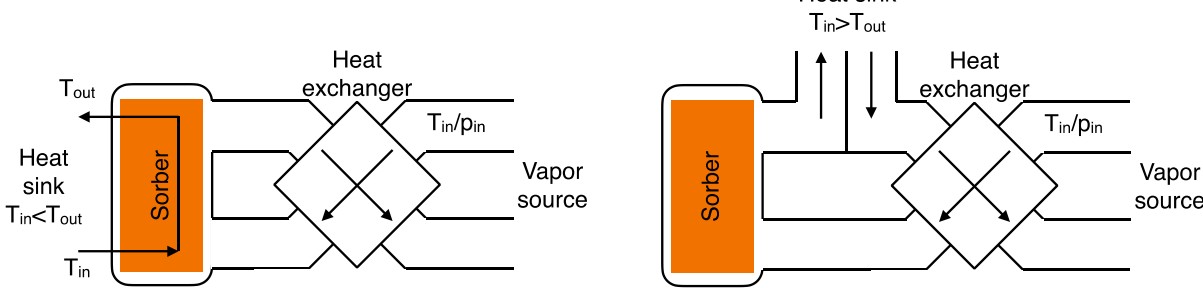

**Figure 3.** Illustration of the open fixed process with air to air heat exchanger and separated heat release, liquid on the **left** and air-bound on the **right**.

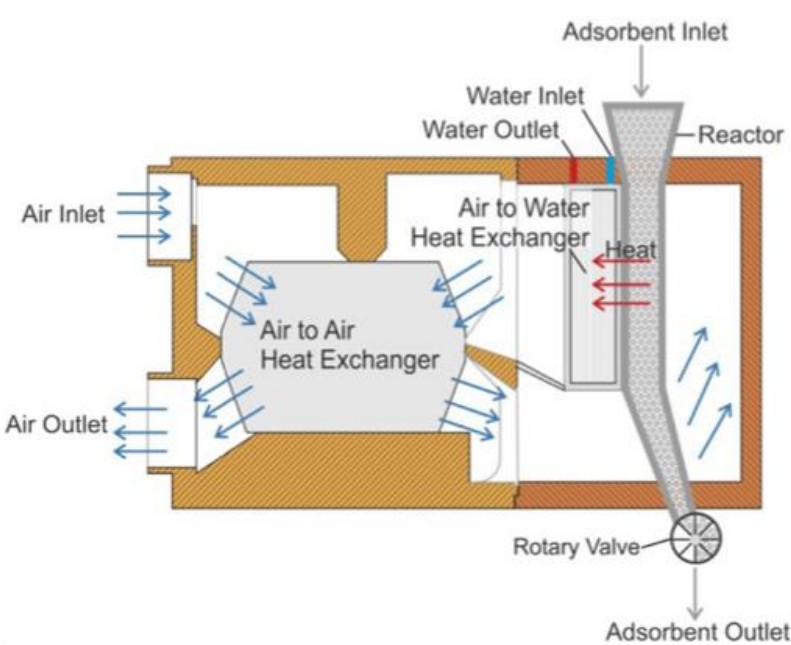

**Figure 4.** Example of a transported open system with zeolite [10].

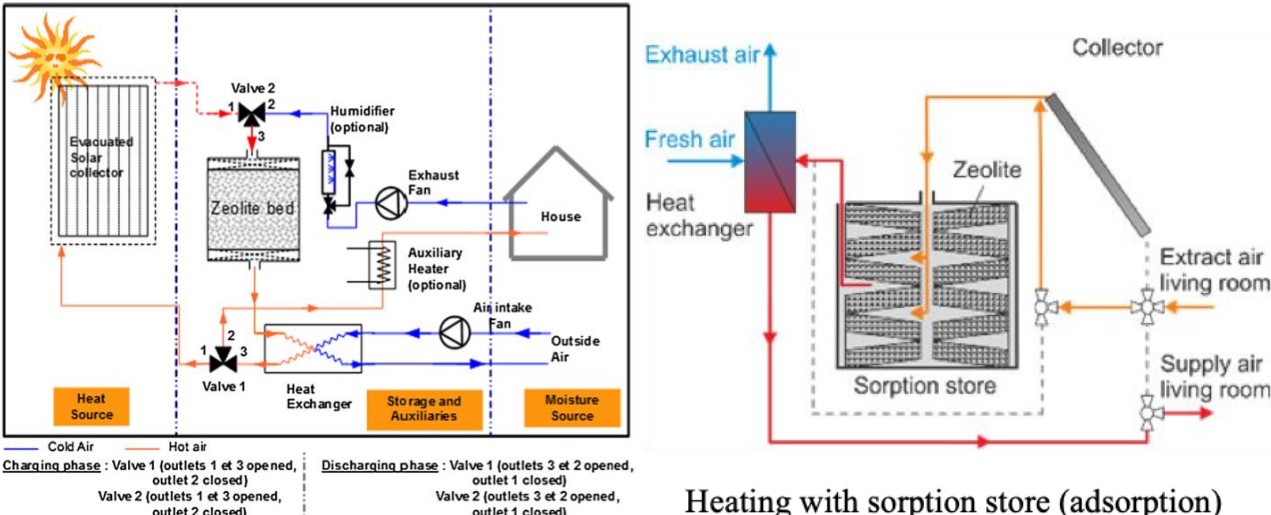

**Figure 5.** Building heat recovery system. Left side illustration from Tatsidjodoung et al. [13] Right side illustration from Weber et al. [14].

As can be seen by the uniform boundary parameters in the process illustrations in Figure 2, variance in fixed to transported process does not exert influence on the governing boundary conditions. However, there is disparity in power dynamics and terminal state. The transported process is characterized by constant power and temperature. Adjustment of all flows provides preferred steady state operation. Contrastingly, the fixed process attributes strong state-of-charge conditioned power and temperature dependence, complicating testing procedure by demanding variant HTF flow and making final point of discharge, a question of minimum acceptable output power, uncertain.

From the description of the sorption heat storage processes, it is recognised that, to fully describe system performance in charging and discharging, sorbent and sorbate temperatures or pressures need to be specified. In the closed system, this translates to eight distinct HTF temperatures. These are desorber inlet and outlet, condenser inlet and outlet, evaporator inlet and outlet, and absorber inlet and outlet. In the open process, water

vapor is released and sourced from the ambient air, and evaporator and condenser inlet and outlet temperatures are extended with sorbate (water vapor) pressure.

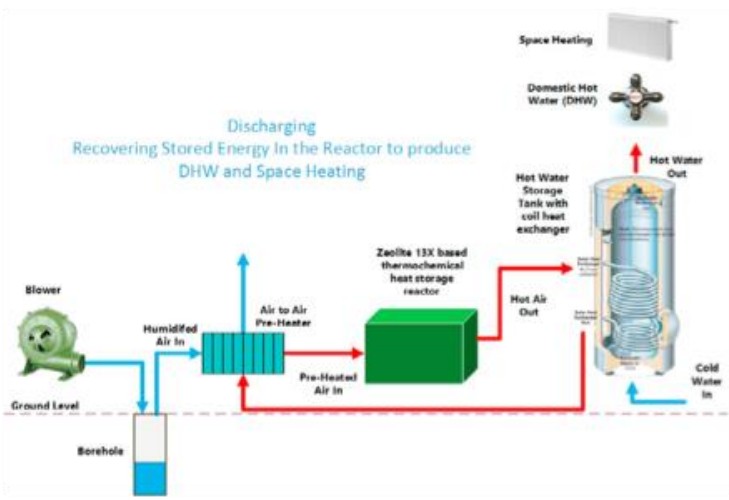

**Figure 6.** Illustration from Gaeini et al. [15]. Energy is sourced from the borehole.

## 3. Reporting in Literature

Looking into the literature, it is found that up to now, and particularly in review papers adequate declarations of operating temperatures are often missing. For example, Krese et al. [16] and Solé et al. [17] list charging and discharging temperatures without stating condensation and evaporation conditions. Review papers from Cabeza et al. [9], Palomba and Frazzica [18], Tatsidjodoung et al. [19], and Yu et al. [20] resort to the term 'operation conditions' in their sorption materials and energy density inventory, often reducing the data to a single charging temperature. A commonly used illustration is shown in Figure 7, plotting energy density based on a single temperature [20]. This is misleading as it does not accurately show the specificities of sorption processes and their dependence on various temperatures and pressures. This is problematic for sound comparison, since, as clearly shown, the energy density of a sorbent material is far from dependent on only one temperature.

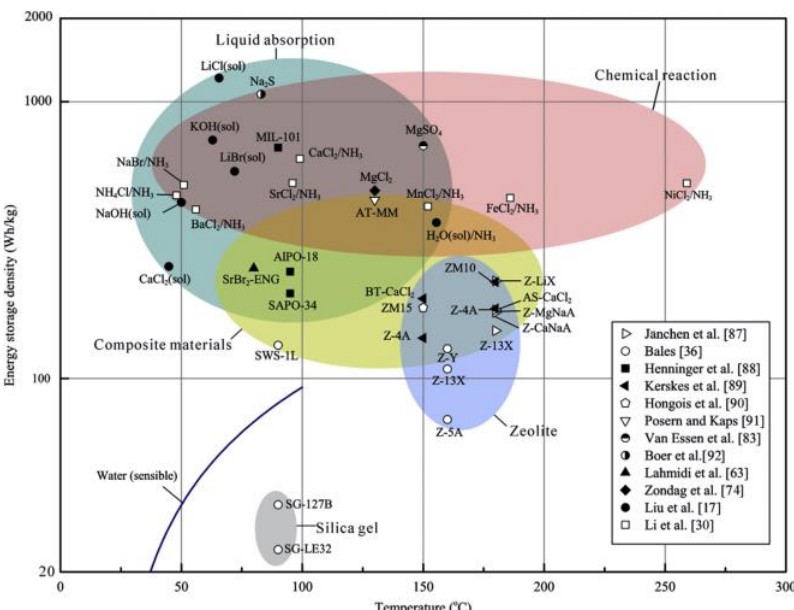

**Figure 7.** Illustrations showing energy density based on a single temperature [20].

In their extensive salt hydrate study, Donkers et al. [21] present a diagram showing energy density vs. desorption and sorption temperature conditions, based on constant evaporator and condenser vapor pressures of 20 and 12 mbar, respectively, equivalent to 18 °C and 10 °C evaporating temperature, as can be derived from the water vapor curve in Figure 2. The diagram is shown in Figure 8. This is a step towards founded comparison, even though non-constant desorption and sorption temperatures prevent strict balancing.

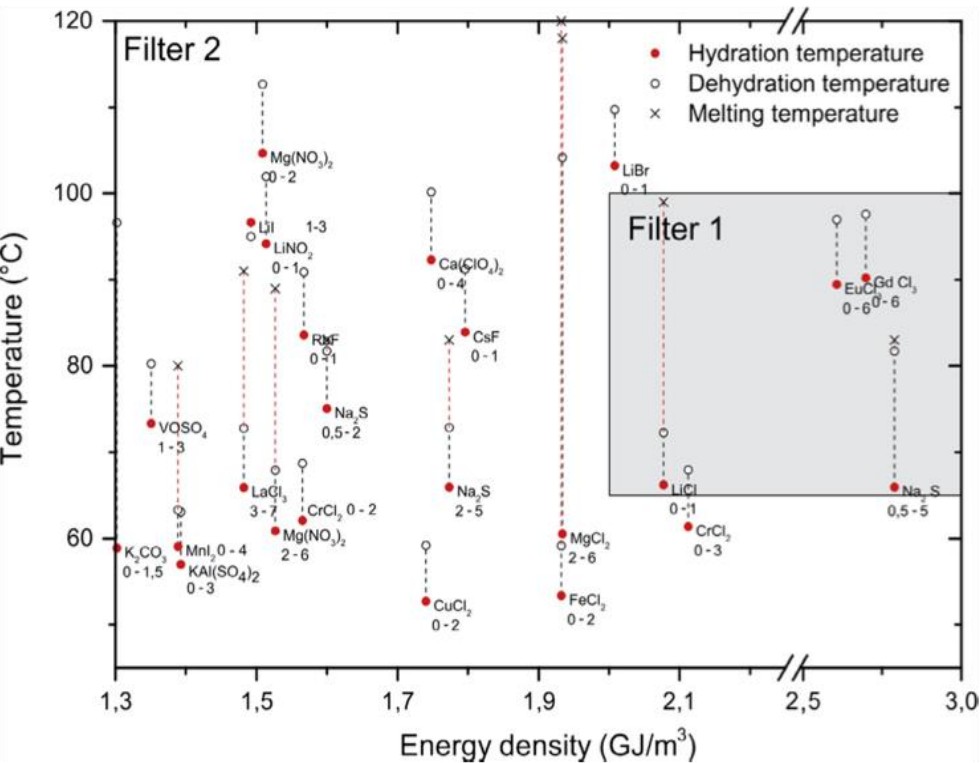

**Figure 8.** Energy density vs. desorption and sorption temperature at constant condenser and evaporator temperature of 20 mbar and 12 mbar respectively by Donkers et al. [21].

In their review on liquid absorption heat storage, Mehari et al. [22] have taken the listing of boundary conditions a step further to include charging (desorption), discharging (sorption) and evaporation temperature, nevertheless not including condensation temperature.

At this stage it is important to mention that it is not the authors' intention to criticize the mentioned works, but rather to point to the fact that insufficient regard has been granted to the need for complete declaration of testing temperatures for the sake of reproducibility of experiments and performance comparison.

While a general absence of clear declaration of testing conditions is found in review papers, researchers struggle to identify proper boundary conditions for sorption heat storage in domestic application. Table 1 shows a sample list of testing conditions from related research papers. Caution is called-for when interpreting the sorption temperatures. While condenser and evaporator temperatures are fixed and desorption temperature is generally the final temperature reached, sorption temperature conditions are not uniform, and refer either to the maximum sorbent temperature reached without load, the required output temperature whereby discharging proceeds until this temperature falls, or the input temperature whereby discharging is carried out until temperature increase to the output temperature ceases. Selection often depends on development level, material, component or system. Suggested energy densities are deliberately not included in the list. The considerable variation of all temperature conditions and temperature uncertainties makes comparison of energy density misleading. Consequently, clear evaluation of progress in the development of sorption heat storage for building application cannot be attained.

**Table 1.** List of testing temperatures from research papers. Red values are out of bound with reference to the proposed testing temperatures in Table 2.

| Authors | Temperature (Humidity) Conditions | | | | Ref. |
|---|---|---|---|---|---|
| | **Desorption** | **Condensation** | **Evaporation** | **Sorption** | |
| Liu et al. | 45–155 °C | 30 °C | 10 °C | 20 °C | [23] |
| Henninger et al. | 140 °C | 35 °C (5.6 kPa) | 10 °C (1.25 kPa) | 30 °C | [24] |
| Henninger et al. | 95 °C | 35 °C (5.6 kPa) | 10 °C (1.25 kPa) | 40 °C | [24] |
| Jeremias et al. | 140 °C | 35 °C (5.6 kPa) | 35 °C (5.6 kPa) | 40 °C | [25] |
| Gaeini et al. | 150 °C | - | 10 °C (1.25 kPa) | 20 °C | [26] |
| Courbon et al. | 80 °C | 10 °C (1.25 kPa) | 10 °C (1.25 kPa) | 30 °C | [27] |
| Fröhlich et al. | 140 °C | 35 °C (5.6 kPa) | 10 °C (1.25 kPa) | 40 °C | [28] |
| Fröhlich et al. | 120 °C | 10 °C (1.25 kPa) | 10 °C (1.25 kPa) | 20 °C | [28] |
| Tohidi et al. | - | - | 21.4 °C (25 °C 80%RH) | 25 °C | [29] |
| Ponomarenko et al. | - | - | 28 °C (3.7kPa) | 50 °C | [30] |
| Zhu et al. | - | - | 26 °C (30 °C 80%RH) | 30 °C | [31] |
| Ristić et al. | 150 °C | 35 °C (5.6 kPa) | 10 °C (1.25 kPa) | 25 °C | [32] |
| Jabbari-Hichri et al. | 150 °C | 3 °C (0.78 kPa) | 3 °C (0.78 kPa) | 20 °C | [33] |
| Casey et al. | 90 °C | 0% RH | 13 °C (14 °C 95%RH) | 14 °C | [34] |
| Zhang et al. | 110 °C | - | 12–17 °C (20 °C 60–80%RH) | 20 °C | [35] |
| Posern et al. | 130 °C | - | 31 °C (35 °C 80% RH) | 35 °C | [36] |
| Permyakova et al. | 80 °C | 10 °C (1.25 kPa) | 10 °C (1.25 kPa) | 30 °C | [37] |
| Stritihd and Bombac | 95 °C | 22 °C | 18 °C | 22 °C | [38] |
| Lehmann et al. | 180 °C | 7 °C (1.0 kPa) | 7 °C (1.0 kPa) | 20 °C | [39] |
| Jänchen et al. | 450 °C | 4.5 °C (0.85 kPa) | 4.5 °C (0.85 kPa) | 22.5 °C | [40] |
| Donkers et al. | 120 °C | 17.5 °C (2.0 kPa) | 10 °C (1.25 kPa) | 65 °C | [21] |
| Donkers et al. | 100 °C | 17.5 °C (2.0 kPa) | 10 °C (1.25 kPa) | 50 °C | [21] |
| Fumey et al. | 55–65 °C | 20 °C | 20 °C | 28 °C | [41] |

**Table 1.** *Cont.*

| Authors | Temperature (Humidity) Conditions | | | | Ref. |
|---|---|---|---|---|---|
| | **Desorption** | **Condensation** | **Evaporation** | **Sorption** | |
| Johannes et al. | 180 °C | - | 14 °C (20 °C 70%RH) | 20 °C | [42] |
| Tatsidjodoung et al. | 180 °C | - | 14 °C (20 °C 70%RH) | 20 °C | [13] |
| Michel et al. | 82 | 20 °C | 6 °C | 25 °C | [43] |
| Weber et al. | 180 °C | 8 °C (1.0 kPa) | 8 °C (1.0 kPa) | 20 °C | [14] |
| Aydin et al. | 80 °C | −40 °C (0.018 kPa) | 19 °C (20 °C, 2.16 kPa) | 20 °C | [44] |
| Gaeini et al. | 190 °C | - | 8 °C (10 °C, 90%RH) | 10 °C | [15] |
| Nonnen et al. | 180 °C | 8 °C (1.0 kPa) | 13 °C (28 °C, 1.5 kPa) | 28 °C | [45] |
| Finck et al. | 103 °C | 20 °C | 15 °C | 20 °C | [46] |
| Köll et al. | 180 °C | 17 °C | 20 °C | 20 °C | [47] |
| Palomba et al. | 90 °C | 30 °C | 10 °C | 35 °C | [48] |
| Brancato et al. | 90 °C | 30 °C | 12.5 °C | 37 °C | [49] |
| Zhao et al. | 85 °C | 18 °C | 30 °C | 40 °C | [50] |
| Jiang et al. | 150 °C | 15 °C | 15 °C | 30 °C | [51] |
| Zhang et al. | 72 °C | 20 °C | 12 °C | 38 °C | [52] |
| Le Pierrès et al. | 59 °C | 16 °C | 15 °C | 27 °C | [53] |

This short excursion into literature shows that, in the past, insufficient attention has been given to uniform material testing temperatures and even less to distinct HTF temperatures when evaluating performance for the building application. Consequently, comparison between tested materials and systems is problematic and progress evaluation challenging. This circumstance can only be overcome by finding uniformity in testing.

The question of realistic temperatures for sorption heat storage systems for building applications has been given thought to by many authors. Courbon et al. [27] suggested a sorption temperature of 30 °C, a desorption temperature of 80 °C and a vapor pressure of 12.5 mbar equivalent to an evaporating temperature of 10 °C for both sorption and desorption. They stated that solar thermal charging at temperatures above 100 °C are not practicable. Palomba and Frazzica [18] and again Frazzica et al. [54], point to the huge inhomogeneity among prototype testing results and call for the definition of common testing methods and key performance indicators in order to make the prototype characterization comparable. Scapino et al. [55] stated that, to make research on sorption heat storage comparable, common key performance indicators should be adopted by the research community, emphasizing the need for common reference temperatures. Again, Courbon et al. [56] stated that despite the fact that numerous studies have focused on $CaCl_2$-based composites, comparison is not possible, since testing temperatures vary strongly and are often out of range for domestic application. Fumey et al. [12] proposed a method termed temperature effectiveness for evaluation of material specific component design, yet clear system comparison is not reached and consequently progress evaluation unattained. A step towards sound evaluation of materials, components and systems is suggested by Hauer et al. [57] based on a four static temperature testing approach. Nevertheless, they do not provide clearly defined temperatures and as shown in Figure 1 and

the accompanied discussion, in discharging, both sorbent maximum and minimum temperatures are important, amounting to five temperatures. Additionally, their proposition focuses on the material level and not the system, thus excluding HTF temperatures. HTF temperatures are considered by Frazzica et al. [54]. In their unified methology to sorption heat storage testing, they clearly point to the need for eight individual HTF temperatures. They propose simulation based performance evaluation founded on dynamic temperature profiles. This is well fitting for the evaluation of a system in a specific application, but basis on clear system characterization for simulation validation. Often sorption heat storage research is not at this advanced technology readiness level.

For this reason, we suggest that a simple mutually agreeable testing outline for clear comparison, focusing on realistic static operating temperatures is required. This guideline is able to grant comparison between materials, components and systems from an early stage of development on.

Two questions emerge: Is it possible to define a singular set of static conditions to cover all four system processes? And, what are viable specific conditions for the building application?

## 4. Realistic Testing Conditions for the Building Application

Apart from the sorbent material dependence, state-of-charge and energy density is directly dependent on the temperature difference between the heat source and sink, and the time provided to approach sorbent equilibrium condition. The latter is a question of material and heat and mass exchanger design. Accepting ideal design conditions, performance is directly bound by the application relevant temperatures, determining power and energy performance.

In the domestic application, the sorption heat storage technology is allocated between sensible heat storage and heat pump, and must accordingly correlate to respective testing standards and operation limits. In this work, European standards are consulted, others are expected to vary only slightly.

The European standard EN 14511 [58] specifies the test conditions for electrically driven compressor heat pumps, for space heating and cooling, well-fitting to this work. Standard heat pump testing conditions for space heating with water based HTF are, 10 °C heat source input and 7 °C output and 30 °C heat sink input and 35 °C output. To extend the description for open systems, a water vapor pressure of 0.87 kPa is fitting. This is the equivalent of 5 °C evaporating temperature, accounting for temperature drop and non-saturation.

Two heat sources are practicable for charging, i.e., solar thermal and electric (renewable). According to the EN 12897 standard [59], specification for closed storage water heaters, the maximum temperature of a domestic solar thermal system is limited to 95 °C. Definition of a temperature limit for electric resistive heating is sorption material bound and not further considered in this study. HTF input and output temperatures in charging are thus taken at 95 °C and 92 °C, following the discharging heat source decrease of 3K. Temperatures for heat release in charging are provided by the EN 14511 standard water to water space cooling guideline. HTF rating for heat release is 30 °C input and 35 °C output. In order to extend the description to open systems, a vapor pressure of 3.0 kPa is fitting, the equivalent of 24 °C evaporating temperature. Tests are performed at the HTF (liquid and air) flow rate obtained from the corresponding standard rating temperature conditions. Table 2 shows the accumulated static temperatures for sorption heat storage testing.

**Table 2.** Temperature guideline for uniform sorption thermal storage testing for space heating in buildings.

| Process | Input Temperature (Vapor Pressure *) | Output Temperature |
|---|---|---|
| Desorption | 95 °C (3.0 kPa) | 92 °C |
| Condensation | 30 °C | 35 °C |
| Evaporation | 10 °C (0.87 kPa) | 7 °C |
| Sorption | 30 °C | 35 °C |

\* Vapor pressure is relevant only for open systems.

It is clear that evaporating temperatures greater than 10 °C and condensing temperatures lower than 30 °C may be encountered under favourable discharging or charging conditions, however, declaration of performance; temperature, power and energy density, at such optimal temperatures may be misleading.

## 5. Discussion and Conclusions

Sorption heat storage performance highly depends on operating temperatures, a factor oftentimes disregarded in review papers and insufficiently regarded within the research community. Figure 1 and the accompanied text explains the required process temperatures and Table 1 provides a glimpse of the great dispersion of operating temperatures for the building application reported in literature. This is an issue also pointed to by authors such as Courbon et al. [27], Palomba and Frazzica [18] and Scapino et al. [55], Hauer et al. [57] and Frazzica et al. [54]. In this proposition for uniform testing conditions with a cross-comparison of current practice reported in literature, effort has been undertaken to show that a single static guideline is applicable for all sorption process types and able to provide simple remedy to this dilemma. Closed systems require only the defined temperatures as provided in Table 2, open systems require vapor pressure in substitute for the condenser and evaporator temperatures. In desorption input temperature (95 °C) and vapor pressure (3.0 kPa) as well as output temperature 92 °C are relevant and condenser temperatures are omitted. In sorption, as indicated, there are slight variations. Systems with air bound heat transport as shown in Figures 2 and 3 right, require evaporation input temperature (10 °C) and vapor pressure (0.87 kPa) as well as sorption output temperature 35 °C, referring to the air temperature. Systems with liquid HTF as illustrated in Figure 3 left, additionally require sorption input temperature 30 °C, with both sorbent temperatures referring the liquid HTF. In the transported process the static temperature conditions can be adjusted, in the fixed process, fluctuation may occur, due to power and state-of-charge correlation. This is little avertable since power is system dependent and cannot be declared for uniform testing. Mitigation can be provided by giving flexibility to the output temperature, lower in desorption and evaporation and greater in condensation and sorption. For this reason, fixed processes reach only partial discharge to input temperature level, the degree of discharge is lower than in transported processes.

The guideline describes realistic, specific conditions encountered at the system-building interface and must not be confused with temperature conditions at the material level. The temperature difference between the HTF temperature and the material temperature depends on the component properties and design. Material temperatures are not further considered in this study. It is clear that from a materials perspective the charging temperature is below 92 °C, the condensing temperature above 35 °C, the evaporating temperature below 7 °C and the discharging temperature above 35 °C. The maximum charging temperature lift is lower than 57 K and the required discharging lift greater than 28K. Material testing in-light-of building application should consider this. To supply domestic hot water at 65 °C as declared by the CEN/TR 16355 recommendation [60], would require a discharging temperature lift greater than 58 K. This cannot be reached without additional boost, the reason why this operation mode has not been considered further in this guideline.

In Table 1 red numbers are out of bounds in accordance to the temperature guidelines of Table 2. Greatest difference is in the evaporation temperature, many authors reported

testing at 10 °C neglecting the heat exchange-induced temperature drop. Sorption (heating) temperature is largely below the required 35 °C and condensing temperatures are generally taken lower than realistic. Materials requiring high charging temperatures will have to resort to resistive heating.

The proposed set of static testing temperatures and pressures is based on conditions that sorption heat storage systems will encounter on their certification path to market. Dynamic behaviour is not included in this guideline. The guideline is realistic, uniform and tough, and will leave many materials and demonstrators struggling to master. With it, materials can be uniformly evaluated in respect to material specific energy density and components and systems can be tested and compared in respect to power and energy density performance. This sets the basis for clear progress evaluation in sorption heat storage research and development. As such, it represents a good reference base allowing an informed discussion on future energy system design. This includes indication of extra measures taken to circumvent restrictive temperature boundaries as proposed here. Examples could be resistive heating to reach higher material temperatures in charging or combinations of sorption storages with electric heat pumps [61] to leverage the overall available temperature lift in discharging, thus allowing as well for production of domestic hot water. Other measures such as integration of solar collectors along with ground heat exchangers to achieve higher source temperatures in discharging can be discussed with reference to the temperature guidelines proposed here.

## 6. Outlook

Sorption heat storage systems are proposed for long-term heat storage for space heating in buildings. To date there are no commercial systems available. While progress in the research community has been made, there is difficulty in clear evaluation on the complete scale from material to components and systems. If the proposed temperature-based testing guideline were to be followed by researchers in the field, it would facilitate performance and progress comparisons and in this way accelerate the development in the field towards the increased use of renewable energy for building space heating. While the suggested testing conditions are only meant to be a guideline, they further enable a critical discussion of over-all system designs. If e.g., in the building context, higher desorption temperatures are being used than what is suggested in the testing guidelines, researchers are welcome to explain required measures on the system level to reach these temperatures. By this, the dedicated sorption research will be automatically pushed towards more realistic and representative system design activities.

**Author Contributions:** Conceptualization, B.F. and L.B.; methodology, B.F.; investigation, B.F.; resources, B.F.; data curation, B.F.; writing—original draft preparation, B.F.; writing—review and editing, B.F. and L.B.; visualization, B.F.; supervision, L.B.; project administration, L.B.; funding acquisition, L.B. Both authors have read and agreed to the published version of the manuscript.

**Funding:** This work was supported by the Swiss Innovation Agency Innosuisse in the frame of the Swiss Competence Centre for Energy Research Heat and Electricity Storage (SCCER HaE), grant Nr. 1155002545 and the Swiss Federal Office of Energy SFOE grant Nr. SI/501605-01.

**Institutional Review Board Statement:** Not applicable.

**Informed Consent Statement:** Not applicable.

**Acknowledgments:** We acknowledge our colleagues participating in the IEA SHC Task58/ECES Annex 33 for the many fruitful discussions we had together and their inputs about performance evaluation and material testing of sorption storages.

**Conflicts of Interest:** The authors declare no conflict of interest.

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
