# Peer review of "Static Temperature Guideline for Comparative Testing of Sorption Heat Storage Systems for Building Application"

_energies, doi:10.3390/en14133754_

Round 1

Reviewer 1 Report

This paper tries to review the literature on the dependency of sorption storage system performance on operating temperatures, and provides some recommendations. However, the main issue is that such an analysis needs a systematic literature review method, and the authors should look at the literature on conducting such analysis before criticizing and reviewing other papers. During this process, you will learn to include - for instance - the search terms to find the right papers, the total number of papers, the strategy to focus on a few papers, how to classify them, etc.  

One minor point on the writing style: I advice using more modest terms and not using the terms such as "neglect", "misleading" or "false impression" while citing the work of other researchers. Each published research paper has its own merits and has been reviewed by other scholars. 

Author Response

Dear Reviewer, many thanks for your input. This paper does not attempt to find evidence for the dependency of sorption system performance on operating temperatures by means of literature review. I am sure you are aware that this dependency is a fact. We present work reported by various researchers and take a closer look at the testing conditions, i.e. temperatures applied and how they are being communicated. The paper has been adjusted to make this more clear. Looking into literature shows that this fact has not been given sufficient attention, a recognition backed by other authors. Showing the operational dependency and pointing to the neglect in literature justifies the definition of static temperatures for comparative testing for mutual system and technology progress evaluation.
We have adapted to more modest terms in the text. Nevertheless, it is generally best to call things by their proper names, point them out and gain understanding. We expect other authors to do the same with our publications.

Reviewer 2 Report

This review included four types of sorption heat storage system and addressed an asserted issue that testing temperatures reported in literature varied strongly, often outside of application scope and frequently even incomplete. Further, they proposed a static temperature and vapor pressure based testing guideline for building integrated compact heat storage systems. This paper is considered for possible publication after some major revisions.

  1. All figures and tables in this review are captured from other literatures. It is suggested to draw a synthetical figure by yourself to illustrate the developing course of the sorption heat storage system and the existing problems.
  2. This review lacks the operation principle of the sorption heat storage system to help readers to understand the importance of the operation temperatures.
  3. It is suggested to introduce Fig. 1 at the beginning in detail rather than in line 135-173.
  4. The authors stated that the sorption heat storage system performance is in respect to not only one temperature. Please explain the relation between the performance and the temperatures physically.
  5. In the caption of Fig. 3, I think “20mbar” and “12mbar” are not correct.
  6. The authors stated “vapor pressure of 12.5mbar equivalent to an evaporating temperature of 10°C” in line 114. Please explain the relation between the vapor pressure and the evaporating temperature physically.
  7. It is suggested to propose the research outlook of the sorption heat storage system in the last part.

Author Response

Dear reviewer, many thanks for your valuable input. Please find our response below your individual noted points.

  • All figures and tables in this review are captured from other literatures. It is suggested to draw a synthetical figure by yourself to illustrate the developing course of the sorption heat storage system and the existing problems.

This request is not clear to us. Figure 1-3 are our own representations, as are Table 1 and 2. A new figure 1 has been added to clarify the required testing temperatures. In the introduction, text has been added, pointing to the recognized need of simple realistic testing procedures.

  • This review lacks the operation principle of the sorption heat storage system to help readers to understand the importance of the operation temperatures.

This has been amended by adding the vapor pressure vs. temperature diagram in figure 1 and adding explanatory text.

  • It is suggested to introduce Fig. 1 at the beginning in detail rather than in line 135-173.

This suggestion has been followed and headings have been renamed.

  • The authors stated that the sorption heat storage system performance is in respect to not only one temperature. Please explain the relation between the performance and the temperatures physically.

This is done with figure 1 and the adhering text.

  • In the caption of Fig. 3, I think “20mbar” and “12mbar” are not correct.

In the referred paper, the caption to figure 5, the figure reprinted in figure 8 (old figure 3) of this paper, states:

Fig. 5. A selection of the database, which fits the drafted working conditions of filter 1 and/or 2, where the gray shaded area fits the working conditions of filter 1. The maximum hydration and the minimum dehydration temperature of the different hydrate couples are plotted against the reaction energy density on material level (open system). The vapor pressure is equal to 20 mbar and 12 mbar during dehydration and hydration, respectively. In addition the lowest melting temperature of the involved hydrates within the reaction is plotted.

  • The authors stated “vapor pressure of 12.5mbar equivalent to an evaporating temperature of 10°C” in line 114. Please explain the relation between the vapor pressure and the evaporating temperature physically.

Reference to figure 1 is made. In figure 1 the relation of temperature to water vapor pressure is given.

  • It is suggested to propose the research outlook of the sorption heat storage system in the last part.

Added an Outlook

Reviewer 3 Report

The article deals with an important and current problem of the research area which is the possibility of using new thermal energy storage technologies for housing. The identified need for unification of research conditions of the solutions tested by researchers seems to be very necessary. The paper is worth publishing, but I recommend to reconsider the manuscript structure. In particular, I suggest considering the following issues:
- In my opinion the title of the paper is too wide. The subject of the article concerns sorption heat storage systems for space heating and domestic hot water.
- There are unfortunate factual errors in the content, e.g.: lines 21-25. The authors suggest that the only way to increase the use of renewable energy resources for space heating and domestic hot water is long-term storage. Maybe it is a key factor but not the only one possibility. In the next sentence they write that such systems have low numbers of charging and discharging cycles. But then they give an example of a solution that does not have this feature. These sentences need to be reconsidered.  
- The authors devote too much attention to criticizing the research done so far by other authors without taking into account their context. 
- The authors provide conditions for testing of the heat storage system. Please consider changing the structure of the article, in which the authors would first present the different approaches of the authors of the topic, then point out the observed difficulties in comparing the obtained results. The result of this analysis would be the author's concept of testing conditions of sorption heat storage systems for space heating and domestic hot water. 

Author Response

Dear Reviewer, please find our answers to your points of concern below:

  • In my opinion the title of the paper is too wide. The subject of the article concerns sorption heat storage systems for space heating and domestic hot water.

Changed title to: Static temperature guideline for comparative testing of sorption heat storage systems for building application

  • There are unfortunate factual errors in the content, e.g.: lines 21-25. The authors suggest that the only way to increase the use of renewable energy resources for space heating and domestic hot water is long-term storage. Maybe it is a key factor but not the only one possibility. In the next sentence they write that such systems have low numbers of charging and discharging cycles. But then they give an example of a solution that does not have this feature. These sentences need to be reconsidered.

Made changes as requested.

  • The authors devote too much attention to criticizing the research done so far by other authors without taking into account their context.

It is not our intention to criticize, but to point out that practice up to now has been unprecise, emphasizing the need for the proposed guideline. The text has been adapted accordingly. Nevertheless, it is not possible to ignore the fact that a declaration of energy density must be accompanied by the complete set of testing temperatures.

  • The authors provide conditions for testing of the heat storage system. Please consider changing the structure of the article, in which the authors would first present the different approaches of the authors of the topic, then point out the observed difficulties in comparing the obtained results. The result of this analysis would be the author's concept of testing conditions of sorption heat storage systems for space heating and domestic hot water.

The manuscript has been heavily restructured.

The introduction points more to the need. Section 2 describes the sorption and system operation. Section 3 reviews practice in literature and Section 4 proposes the temperature guideline.

Reviewer 4 Report

The paper appropriately summarizes the main processes and existing solutions involved in sorption heat storage to improve the overall energetic performance of buildings. It is based on sound scientific papers and the whole structure of the writing is adequate for a scientific journal such as Energies. 

The English language of the paper is good, there are no several flaws in the writing and the readability of the text is high.

Author Response

Dear Reviewer, we appreciate your positive feedback, supporting our research.

Round 2

Reviewer 2 Report

In Fig. 8 caption, "evaporator temperature" should be "vapor pressure", I think.

Reviewer 3 Report

I have no further comments.